# Research on Restoration of Heavy Metal Contaminated Farmland Based on Restoration Ecological Compensation Mechanism

Zheng Cai 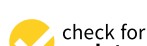 and Xiuli Yang *

School of Public Administration and Law, Northeast Agricultural University, Harbin 150030, China
* Correspondence: xiuliyang@neau.edu.cn

**Abstract:** In recent years, the development of industrialization has led to heavy metal pollution in many agricultural areas in China. The excessive heavy metals in farmland not only affect the normal growth of crops, but also do great harm to human health, which seriously restricts the development of ecology and food health in China. In order to improve the problem of heavy metal pollution in rural areas, the current situation of heavy metal pollution in rural areas is analyzed based on the innovative ecological compensation mechanism for remediation, and the external theory, public goods theory and other relevant theories are combined to obtain the ecological compensation strategy for heavy metal farmland soil remediation, and on this basis, the basic framework of ecological compensation for metal farmland remediation is constructed. Finally, effective environmental treatment suggestions are put forward according to the development requirements of ecological compensation in heavy metal farmland areas. The case study shows that different ecological restoration schemes have been adopted for a heavy metal farmland pollution area, and the environment has been improved according to the ecological compensation scheme. The total amount of ecological compensation for heavy metal farmland is CNY 32.35 million, of which the cost of seriously polluted farmland is the highest among the environmental values, with the cost of restoration per acre of CNY 65,000, indicating that the heavy metal areas are more expensive and have more obvious impact on the environment. The research content has important reference significance for the ecological environment treatment of heavy metal farmland pollution areas in China.

**Keywords:** heavy metal pollution; ecological compensation mechanism; soil remediation; governance opinions

## 1. Introduction

The development of industrialization has promoted the rapid development of social employment and economy. However, the environmental impact caused by industrialized economy in the development period cannot be ignored, especially the heavy metal pollution in farmland, which seriously affects the social health and the stable development of agriculture. People depend on food, which shows that agriculture is related to people's health and social stability. Agricultural fields in many rural areas are polluted by heavy metals due to the random discharge of industrial wastes. According to relevant data reports, heavy metal pollution causes grain production reduction of 12 million tons in China every year, and direct agricultural economic loss is up to CNY 30 billion (indirect economic losses such as ecological restoration costs and social impact costs were not taken into account) [1,2]. It is therefore imperative to strengthen heavy metal farmland environmental governance, and it is necessary for all departments in our country to strengthen environmental ecological governance to ensure agriculture and food safety. The main cause of heavy metal pollution in farmland is that human beings ignore ecological protection in the development of economic activities [3,4].

According to the experience of ecological protection at home and abroad, it is found that harnessing rural ecological problems based on the ecological compensation mechanism is an effective way to improve the farmland environment and increase agricultural crop yield. At the same time, due to the excessive concentration of metal elements in metal-polluted soil, the farmland soil has problems of drought and high salinity, which inhibit the growth of plants. At present, the main remediation technologies of heavy metal contaminated soil are physical and chemical remediation technology, microbial remediation technology, ecological remediation technology etc. [5].

Therefore, based on the ecological compensation mechanism of heavy metal pollution in farmland, combined with the Nemero index method and the relevant theories of the Soil Environmental Quality Standard, the ecological compensation framework of heavy metal pollution in farmland was constructed in this study, and effective farmland remediation suggestions were put forward according to the characteristics of heavy metal pollution in farmland, so as to improve the heavy metal pollution in farmland and promote the sustainable development of agricultural economy.

## 2. Ecological Compensation Theory Research and Heavy Metal Farmland Pollution Grade Evaluation

### 2.1. Research on Ecological Compensation Theory

Heavy metal pollution of farmland is a common ecological pollution problem of farmland, which is mainly caused by a large number of toxic and harmful heavy metal substances flowing into farmland, destroying the ecological mechanism and structure of farmland soil and causing excessive metal substances in farmland. Common toxic and harmful heavy metals include cadmium (Cd), nickel (Ni), copper (Cu), zinc (Zn), chromium (Cr), arsenic (As), mercury (Hg) etc. Cadmium metal pollution is especially particularly serious [6,7].

Industrial manufacturing enterprises discharge some harmful substances exceeding the standard into the farmland environment through water or air, resulting in a series of physical and chemical phenomena in the farmland soil. For example, toxic and harmful heavy metals will change the structure of farmland soil through oxidation, deposition, dissolution, reduction etc. Because different ecological crops have different tolerance to heavy metals, even some metals are difficult to degrade, such as cadmium and nickel, as they will accumulate in the soil for a long time and be absorbed by plants, which will inhibit the photosynthesis, biological enzyme reaction and nutrient absorption of plants, and affect the growth of crops [8]. At the same time, various heavy metal substances of crops grown under heavy metal farmland also seriously exceed the standard, which will cause functional deterioration, poisoning, carcinogenesis and other problems when eaten by people, affecting human health.

### 2.2. Definition of Heavy Metal Farmland Pollution

There are many research methods for investigating heavy metals in farmland. Here, heavy metal pollution in farmland was studied by comparative analysis method. Nerome pollution index and pollution single factor index were used to express the farmland pollution; the Nemero index is a weighted multi-factor environmental quality index that takes into account extreme values or highlights maximum values [9]. Among them, the single factor index of farmland pollution is shown in Equation (1).

$$P_i = \frac{C_i}{S_i} \tag{1}$$

$C_i$ represents the measured concentration value of heavy metals. $P_i$ represents the quality index of heavy metals in soil. $S_i$ represents the evaluation coefficient of heavy metals. In the actual definition of farmland pollution, the polluted farmland is generally expressed as $P_i > 1$, and the non-polluted farmland soil as $P_i \leq 1$. The larger the value of $P_i$, the more serious the heavy metal pollution in farmland [10]. In the definition of heavy

metal pollution in farmland, there is not only one harmful heavy metal in farmland, and some special areas need to consider the influence of multiple heavy metals at the same time. Thus, a single pollution factor is not enough to reflect the actual situation of farmland [11]. Therefore, the Nerome pollution index is used to reflect the comprehensive impact of heavy metals on farmland (2).

$$I = \sqrt{\frac{p_{i\max}^2 + P_{i\text{ave}}^2}{2}} \tag{2}$$

*I* represents the comprehensive pollution index of heavy metal farmland, $p_{i\max}$ represents the largest single pollution index and $p_{i\text{ave}}$ represents the average single pollution index. The Nerome pollution index can more accurately reflect the situation of farmland pollution, and can highlight the main situation of farmland pollution, which is convenient for people to take effective control measures on polluted farmland. The single factor soil pollution evaluation is shown in Table 1 [12].

**Table 1.** Assessment of soil pollution level.

| Pollution Levels | Pollution Status | Pollution Evaluation Index | Farmland Quality |
|---|---|---|---|
| Level one | clean | $p \leq 0.7$ | In safe and healthy condition |
| Secondary | Relatively clean | $0.7 < p \leq 1$ | Relatively safe, but on the verge of pollution |
| Level three | Slight | $1 < p \leq 2$ | Heavy metals in farmland soil obviously exceed the safe value, and plant growth is affected to a certain extent |
| Level 4 | Medium | $2 < p \leq 3$ | Farmland soil, microorganisms, plant growth restriction |
| Fifth grade | Serious | $p > 3$ | Plants die, soil is heavily polluted |

In the evaluation of farmland pollution, generally the main reference is the soil quality label and the environmental background. Different heavy metal-polluted farmlands are different in actual evaluation due to different metal elements and degrees of pollution. Therefore, it is convenient to conduct a unified analysis of polluted farmland across the country, and uniformly adopt the evaluation standard ($S_i$) for soil material pollution in farmland in the "Soil Environmental Quality Standards" [13]. The relationship between soil quality grade and pH (hydrogen ion concentration, PH) is shown in Table 2.

**Table 2.** Soil environmental quality evaluation criteria.

| Heavy Metal | Farmland Type | | Level One | Secondary | Level Three |
|---|---|---|---|---|---|
| | | | Farmland Environment Background | $6.5 \leq \text{pH} \leq 7.5$ | $\text{pH} > 6.5$ |
| Cd | | $\leq$ | 0.20 | 0.3 | 1 |
| Hg | | $\leq$ | 0.15 | 0.5 | 1.5 |
| As | | $\leq$ | 15 | 25 | 30 |
| Cu | Paddy field, farmland, orchard field, dry field | $\leq$ | 35 | 100 | 400 |
| Cr | | $\leq$ | 90 | 300 | 400 |
| Pb | | $\leq$ | 90 | 200 | 300 |
| Zn | | $\leq$ | 100 | 250 | 500 |
| Ni | | $\leq$ | 40 | 50 | 200 |

In the treatment of farmland polluted by heavy metals, the quality of farmland soil should be fully considered, and necessary environmental treatment measures should be taken according to the farmland pollution situation to ensure the healthy development of modern agriculture [14]. The restoration ecological compensation mechanism is to coordinate the relationship between the main body of ecological compensation and the

department on the basis of improving the ecology of farmland, so as to meet the restoration requirements of heavy metal polluted farmland in a specific form. In the study of compensation mechanism for farmland polluted by heavy metals, it is necessary to optimize the interest relationship among enterprises, farmers and the government, and take reasonable and effective measures to ensure the stable development of agricultural economy [15]. Only in this way can heavy metal polluted farmland be treated more effectively and regional economic development status be improved.

## 3. Construction of Ecological Compensation Framework for Heavy Metal Polluted Farmland

### 3.1. Investigation on Compensation of Heavy Metal Polluted Farmland Users

The compensation survey of users in heavy metal polluted farmland areas is mainly carried out in the form of questionnaire surveys, field visits and consulting regional agricultural economic documents. Therefore, 276 heavy metal pollution compensation questionnaires were distributed in the Changsha-Zhuzhou-Xiangtan area, including the economic type of the local people, the understanding of heavy metal pollution, and the compensation provisions for heavy metal pollution. Finally, 265 copies were recovered, with a recovery rate of 96.01%.

According to the investigation of a heavy metal farmland pollution area, people in heavy metal pollution areas mainly rely on agricultural economy. At the same time, many enterprises choose to build factories and carry out production in rural areas. Although it has driven the economic development of the region in the short term, in the long run, the agricultural ecology of some regions has been affected by industrial pollution, resulting in some cultivated farmland becoming heavy metal contaminated farmland [16]. Therefore, it is necessary to investigate the agro-ecological situation in the region and comprehensively understand the situation of heavy metal pollution in farmland and cultivated land in the region, so as to provide effective reference for environmental and ecological governance in agricultural areas [17]. In total, 276 agricultural planting personnel in the area were investigated, accounting for 83.6% of the total sample number. The specific data are shown in Figure 1.

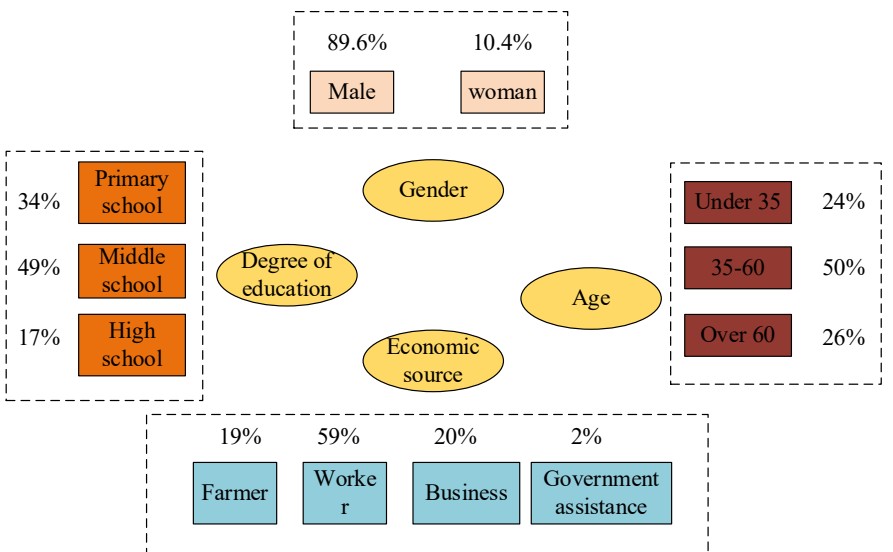

**Figure 1.** Survey data information of agricultural personnel.

From the survey data summary, overall cultural quality of the 276 agricultural growers who participated in the survey is not high: they mainly live on labor and farming [18,19]. In terms of gender data and age, agricultural workers are mainly male, ranging in age from 35 to 60. Through interviews and questionnaires, most respondents noted that farmland had

been polluted, such as the yield and quality of cultivated crops had been affected, while the biomass in farmland had declined sharply, including fish and frogs in rice [20]. However, some people are not aware of the consequences of the pollution and have not experienced adverse effects from daily consumption of crops from contaminated areas. This means that in heavy metal polluted farmland areas, people are aware of farmland ecological environment pollution, but not aware of the impact of ecological pollution. Therefore, the government officials should carry out health and ecological education for the people in the area and explain the related hazards.

### 3.2. Construction of Farmers' Ecological Compensation Intention Model

It is necessary to fully understand the attitude of agricultural personnel towards ecological compensation projects through the investigation of user compensation for personnel in a heavy metal polluted area. Only by identifying relevant influencing factors can we effectively implement farmland ecological compensation mechanism for heavy metal pollution. The main factors affecting the participation of agricultural personnel in ecological compensation include annual income, active labor force, political status, educational level and age. Based on the above factors, this paper constructs an evaluation model of farmers' ecological compensation intention [21]. It is defined $c$ as the model explanatory variable, which indicates the attitude of agricultural personnel towards ecological compensation, where 1 is used for willingness and 0 for non-acceptance. The influencing factors, including annual income, labor force and political status, are all represented by variables $X$, as shown in Equation (3).

$$X = (X_1, X_2, X_3, ..., X_n) \tag{3}$$

In Equation (3), $n$ is the factor serial number. Then the agricultural personnel's attitude towards ecological compensation is as seen in Equation (4).

$$Y = \delta_0 + \delta_1 X_1 + \delta_2 X_2 + \delta_3 X_3, ..., \delta_n X_n \tag{4}$$

$\delta_n$ represents a constant term used to explain some variables. Value is assigned to ecological compensation factors, as shown in Table 3.

**Table 3.** Explanation of variable factor assignment.

| Variable Factors | Factor Name | Assign Value 1 | Assign Value 0 |
|---|---|---|---|
| X1 | Annual income of agricultural personnel | Annual income higher than 30,000 | Annual income less than 30,000 |
| X2 | Homework workforce | More than two people | Less than two people |
| X3 | Political status | Party members and cadres | Non-party members and cadres |
| X4 | Education level | High school or above | Lower than high school |
| X5 | Age | Less than 35 | Greater than 35 |
| X6 | Farmland planting area | More than 5 acres | Less than 5 acres |
| X7 | Farmland pollution survey | Polluted | Unpolluted |
| X8 | Are crops affected | Crop growth is affected | Normal crop growth |
| X9 | Whether to produce crops | Sell crops | Don't sell crops |

When constructing the farmers' intention model, the economic and family conditions of agricultural personnel should be fully considered. While meeting the requirements of local environmental protection, corresponding help should be provided to users with difficulties [22], on the basis of satisfying the agricultural ecological health, sharing the economic burden of agricultural personnel and promoting the treatment of heavy metal pollution in farmland.

### 3.3. Compensation Standard

Heavy metal polluted farmland is mainly affected by cadmium, nickel, copper, zinc, mercury and other harmful substances, which is difficult to control in practice. In addition,

heavy metal pollution in farmland has irreversible impact on ecological restoration, which is difficult to recover only through the natural environment, and the cycle is long [23]. Therefore, it is necessary to clarify the situation of farmland polluted by heavy metals, and implement the corresponding compensation scheme and farmland restoration technology according to the situation of farmland pollution, so as to effectively improve the problem of farmland polluted by heavy metals [24]. With reference to the single-factor soil pollution standard in Table 1 and the Soil Environmental Quality Standard in Table 2, under the condition of considering the factors of compensation, it is necessary to further classify the degree of heavy metal farmland soil pollution and obtain the unit compensation standard of pollution level, as shown in Table 4.

**Table 4.** Division of compensation costs for farmland polluted by heavy metals.

| Classification | Clean | Relatively Clean | Slight | Medium | Serious |
|---|---|---|---|---|---|
| Heavy metal pollution index | $p \leq 0.7$ | $0.7 < p \leq 1$ | $1 < p \leq 2$ | $2 < p \leq 3$ | $p > 3$ |
| Cost (yuan/acre) | 0 | 3050 | 10,050 | 31,000 | 62,000 |

The premise of implementing ecological compensation scheme in rural heavy metal polluted farmland is to meet the standards and requirements of ecological compensation, which should not only meet the requirements of ecological environment management, but also meet the requirements of fairness and justice. Generally speaking, the key to the implementation of the whole ecological compensation mechanism is who causes the pollution, who is responsible for the treatment, and who protects and who benefits [25]. Ecological environment protection has the characteristics of publicity. Every citizen has the obligation to protect it and enjoys the right to use it. The analysis of heavy metal pollution in farmland shows that the direct causes of farmland ecological pollution include the discharge of toxic and harmful substances by enterprises and the destruction of ecological environment by individual users. In the treatment of polluted farmland, environmental protection compensation should be requested from the party that damages the environment, and this part of compensation will become an important source of funds for environmental management, maintenance and protection [26].

Meanwhile, a large number of local people are needed to participate in the treatment of heavy metal polluted farmland to ensure the smooth implementation of ecological protection activities. This includes government leaders, environmental technicians and local people, who need to come together and be involved. We should encourage and support those involved in ecological development and maintenance, establish a corresponding reward mechanism for environmental governance and reflect the principle that those who protect will benefit. The incentive cost of this part should also be borne by ecological vandals, and corresponding compensation should be given to the management of heavy metal farmland.

According to the scheme of farmland ecological treatment of heavy metal pollution, the main body of compensation is the actor who destroys farmland ecology, and the beneficiaries are the relevant personnel involved in environmental protection and management. For the analysis of farmland pollution sources, farmland pollution sources are mainly concentrated in three aspects, as shown in Figure 2 [27].

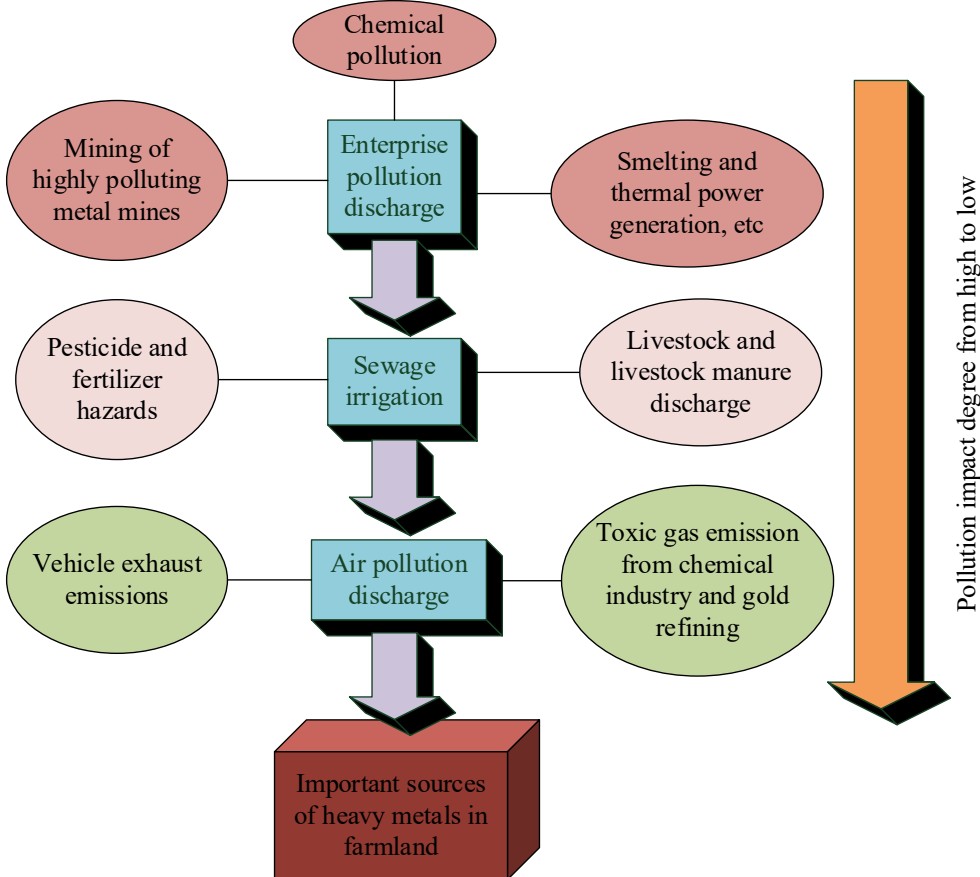

**Figure 2.** Main sources of farmland heavy metals.

Heavy metal pollution in farmland is mainly affected by sewage discharge, sewage irrigation and waste discharge. Among them, sewage discharge from enterprises has the greatest impact, mainly involving chemical industry, mining and other high-risk pollution. Whether the compensation mechanism is effective requires full consideration of these factors. The final environmental compensation standard can be understood as the compensation for environmental governance and crop reduction of landowners, which can be expressed in terms of opportunity value and environmental cost, as shown in Equation (5) [28].

$$C_z = C_c + C_e \tag{5}$$

$C_c$ represents opportunity value and $C_e$ represents environmental cost.

In the formulation of compensation standards, environmental costs are priced through factors such as pollution sources and consumption of environmental resources, which are closely related to social and economic development. However, the socio-economic effects of environmental resources are varied, and some cannot be directly measured in monetary terms, whereas crops grown on farmland can be. Opportunity cost mainly reflects the criteria to evaluate environmental restoration through farmland ecological environment restoration. In the formulation of actual compensation standards, the compensation standards caused by different pollution sources are different. But here we mainly consider the farmland pollution caused by corporate sewage discharge, and mainly refer to the compensation scheme formulation in Table 4 [29]. At the same time, due to the impact of farmland pollution, the harvest of farmland crops is reduced, and the income of farmland

is reduced. Therefore, the opportunity cost of polluted areas is an important content of compensation. The opportunity cost is shown in Equation (6).

$$C_c = \sum_{i=1}^{n} C_i = \overline{C} \cdot T \tag{6}$$

$T$ represents the compensation cycle, $\overline{C}$ represents the average annual loss cost, $C_i$ represents the opportunity cost and $n$ represents the number of days of governance time. In addition, different pollution evaluation levels will lead to differences in pollution control methods and control periods. Therefore, the whole farmland ecological compensation cycle is subject to the farmland environment restoration cycle [30]. According to the relevant standards of environmental governance, compensation is calculated based on the average yield of farmland crops in the 3 years before the reduction. Heavy metal farmland will lead to a certain decrease in crop yield, but the biggest impact is the decline of crop quality. Therefore, according to the relevant national regulations, the heavy metal pollution of farmland will be the corresponding restrictions. In areas polluted by heavy metals, crops will be selected according to the main pollution sources and pollution levels, and agricultural farmers will be compensated according to compensation standards.

### 3.4. Ecological Compensation Scheme for Farmers

According to the different compensation funds, the compensation scheme of heavy metal polluted farmland can be divided into compensation payment and financial payment. Among them, the compensation payment is mainly through the fund payment, but also through the exchange of goods and related supporting services. Common methods include purchasing agricultural products to replace the original farmland crops, setting up rural environmental service stations, etc., which need to be determined according to the actual situation of heavy metal polluted areas [31].

In heavy metal farmland management, enterprises are the main consumers of farmland ecological compensation. Therefore, in the whole ecological compensation, the enterprise is the main body of direct compensation. Meanwhile, it is clearly stipulated in environmental ecological governance in our country that enterprises undertake corresponding environmental governance obligations and fulfill relevant responsibility requirements in the process of social development, and the environmental governance margin paid by enterprises must not be lower than the cost of environmental management [32]. Local governments should force enterprises to pay environmental management fees in accordance with relevant state regulations on environmental health management, and maintain the environment and ecology well in accordance with the law. As the supervisor of environmental compensation projects, the government needs to implement relevant environmental monitoring work and protect the interests of agricultural production victims [33]. In addition, the government sets specific environmental compensation standards according to environmental pollution conditions, and compensates by means of subsidies and cash payments. The entire compensation shall be carried out in accordance with the over refund and less compensation, and the relevant compensation liability system shall be implemented to protect legitimate rights and interests of the victims [34].

The financial payment mainly takes the government as the indirect compensation subject and undertakes the corresponding compensation obligation in the whole environmental compensation work. The government compensation funds mainly come from government financial subsidies and local taxes [35]. Due to the uncertainty and periodicity of the whole heavy metal farmland compensation, government compensation mainly solves the problem of economic development and ensures the stable development of regional economy.

## 4. Case Analysis

### 4.1. Specific Cases

In the process of production and operation, a rural chemical enterprise failed to discharge industrial production sewage into the rural river channel as required. People in this area use the polluted water for irrigation, spraying and fertilization of farmland without knowing, resulting in heavy metal pollution of a large amount of farmland in this area. Soil samples were collected from 265 typical polluted places in the polluted area, impurities were filtered, and the content of heavy metal elements in the soil was detected by atomic fluorescence spectrometry. According to the inspection of the expert group, the contaminated farmland in this area is up to 34,800 acres, including 30,000 acres of contaminated farmland, 3200 acres of relatively clean farmland, 800 acres of Clean farmland, 500 acres of Medium farmland and 300 acres of Serious farmland. It can be seen that the heavy metal pollution of farmland in this area is relatively serious. The experimental analysis of heavy metal property in polluted farmland is shown in Figure 3.

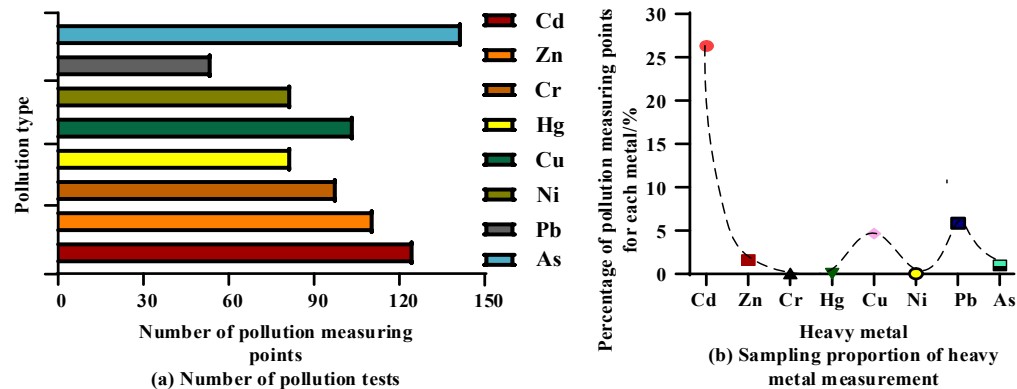

**Figure 3.** Experimental results of heavy metal farmland pollution points.

Figure 3a shows the number of pollution sources at the detection points of polluted farmland. According to the detection results of harmful elements in 150 typical detection sites in the heavy metal polluted farmland, heavy metals are mainly As, Cd, and Zn. As appeared in most of the experimental detection points, the number was 142, followed by Cd, which appeared in 124 test points. In addition, the proportion of toxic and harmful metals such as Zn and Cu is also relatively high. Figure 3b shows the proportion of harmful elements in the test points. It can be seen that the highest proportion of harmful metals is Cd, up to 26.4%, which exceeds the requirements of Soil Environmental Quality Standard in Table 2. The content of Pb and Cu accounts for 6.8% and 5.2%, respectively. In addition, Zn, Cr, Hg, Ni and As are detected at many detection points, but the contents of these heavy metals are relatively low. According to the existing soil quality safety evaluation standards in China, the contents of these heavy metals are close to the soil quality safety level. Therefore, according to the above test results, it is necessary to evaluate the soil contaminated by heavy metals with different pollution degrees and types, and take necessary treatment measures. The main remediation techniques are ecological environment treatment (such as phytoremediation) and physicochemical extraction. Ecological restoration technology requires a long restoration time.

In addition, in some heavily polluted areas, the absorption of heavy metals by plants will be greatly reduced. Thus, for areas where heavy metals exceed the standard seriously, physical and chemical extraction should be used, assisted by ecological restoration, so as to ensure regional ecological stability. In the rest of the region, ecological restoration is the main task. Table 5 shows some phytoremediation of heavy metals. At present, for Cd, Pb, Cu and other heavy metals, using plants, such as nightshade and shepherd's purse, can obtain better heavy metal absorption effect. However, considering the economic development needs of rural farmland, modified wheat, rice and corn should be planted

in low-pollution areas, which can not only absorb certain heavy metals, but also reduce agricultural losses.

**Table 5.** Phytoremediation of heavy metals.

| Heavy Metal | Restoration Plant | Content mg/kg | Estimated Lifespan (Years) |
|---|---|---|---|
| As | Centipede grass | 40 | 3 |
| Cd | Solanum nigrum | 82 | 2 |
| Zn | Sedum | 32 | 1.5 |
| Cu | Indian acrestard greens | 56 | 2.8 |
| Hg | Canada poplar, mangrove | 15 | 2.6 |
| Cr | Jatropha curcas, reeds | 43 | 2.8 |
| Pb | Indian shepherd's purse | 36 | 4.2 |

Figure 4 shows the remediation of heavy metal polluted farmland with improved wheat remediation technology and physicochemical remediation technology in this case. The final remediation cost of polluted farmland with different pollution degrees is shown in Figure 4a. It can be seen that farmlands with different pollution degrees need different costs in the treatment process. Among them, Clean farmland meets China's farmland environmental treatment safety standards, which do not need to repair due to the lightest heavy metal pollution. Relatively clean farmland is polluted by heavy metals to a certain extent, and some ecological restoration measures need to be taken to restore its ecological function. The cost is 3100 yuan/acre. With the increase of pollution degree, the cost of treatment increases rapidly. For example, the restoration cost of light polluted farmland is 12,000 yuan/acre, and the restoration cost of Serious polluted farmland is up to 65,000 yuan/acre. The restoration time of farmlands varies with different pollution levels. As shown in Figure 4b, the longest restoration time is 40 months for Serious polluted farmland, while the shortest is 15 months for Clean polluted farmland. It can be seen that the more seriously polluted the farmland, the longer the restoration cost and restoration cycle, and the greater the negative impact on the ecological environment and agricultural production.

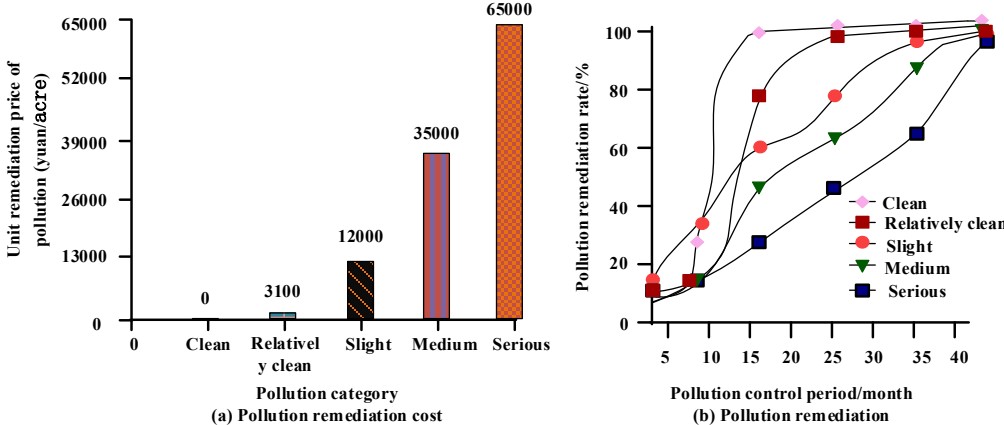

**Figure 4.** Restoration of heavy metal polluted farmland in the case study area.

Table 6 shows the ecological compensation results of heavy metal agricultural areas in this case based on ecological governance and regional agricultural compensation. The final ecological compensation cost reached CNY 32.35 million. In addition to clean farmland, the rest of the heavy metal polluted farmlands need ecological compensation and restoration. Among them, the environmental value cost of Serious polluted farmland is the highest, and the opportunity cost is also the highest because of the most serious ecological damage

caused by pollution to such farmland. In other words, the higher the pollution degree of heavy metal farmland, the higher the resource consumption and repair cost.

**Table 6.** Ecological compensation for heavy metal polluted farmland in the case study area.

| Ecological Compensation Cost | Clean | Relatively Clean | Slight | Medium | Serious |
|---|---|---|---|---|---|
| Polluted area/acre | 0 | 3200 | 800 | 500 | 300 |
| Environmental value (yuan/acre) | 0 | 3100 | 12,000 | 35,000 | 65,000 |
| Opportunity cost | 0 | 1520 | 2150 | 2450 | 3650 |
| Ecological compensation standard (yuan/acre) | 0 | 4550 | 13,200 | 36,000 | 66,000 |
| Ecological compensation amount (million yuan/acre) | 0 | 1300 | 906 | 310 | 124 |
| Final total compensation (million yuan/acre) | | | 3235 | | |

### 4.2. Opinions on the Control of Heavily Polluted Farmland

Metal pollution of farmland is mainly affected by such factors as enterprise pollution discharge, farmland fertilization and industrial exhaust emissions. In particular, the impact of chemical enterprises on the whole agricultural ecology is fatal. In the survey of heavy metal contaminated farmland, most of the farmland was affected by heavy metal contaminated water sources, accounting for 72.63%, followed by pesticides and fertilizers, accounting for 12.62%, and the rest accounted for 14.75%.

According to the survey results, several suggestions are put forward for the remediation of heavy metal farmland. First of all, strengthen regional environmental supervision, establish more effective environmental health supervision regulations, and clarify responsibilities and obligations. We should strengthen institutional supervision and punishment on enterprises in high-pollution areas, and avoid the disorderly disposal of hazardous substances by enterprises from the source. Secondly, do a good job in environmental health control, and take necessary environmental control measures according to the degree of heavy metal pollution in farmland and main pollution elements, including environmental protection publicity, environmental safety education etc. Finally, we should introduce diversified environmental governance means, including ecological environmental governance means, physical governance means and microbial governance means, to improve the quality of regional environment.

### 4.3. Discussion

Heavy metal pollution in rural areas is a hot topic of social concern. Heavy metal pollution has a serious impact on environmental ecology, modern agriculture and food safety. Applying the ecological compensation mechanism to the environmental treatment process of heavy metal polluted areas provides a new direction for the remediation of heavy metal polluted farmland and the development of agricultural economy. Heavy metal pollution control in rural areas has been criticized for a long time [36]. The main reason is that the heavy metal pollution problem in rural areas cannot clarify the responsibility of the pollution subject, the pollution compensation method, the pollution object and the pollution evaluation standard. At the same time, a survey was carried out on the development structure of rural economy in heavy metal pollution areas [37]. Most of the villagers in the remaining villages mainly rely on agricultural economy, while a small number of villagers choose to work in nearby village enterprises. This economic structure led to the local people's lack of awareness of the hazards of heavy metal pollution, and even many people were employed in sewage enterprises, resulting in the phenomenon of shelter and illegal pollution, which affected the restoration of heavy metal contaminated farmland [38].

In recent years, China has strengthened the development requirements for rural environment and food safety, making the agricultural ecological restoration compensation

mechanism gradually perfect. In the ecological compensation mechanism, the main compensation requirement of "who pollutes, who compensates" is clearly stipulated. At the same time, with the gradual improvement of relevant environmental regulatory requirements and the innovation of heavy metal pollution remediation technology, the problem of heavy metal pollution in rural areas has also been alleviated [39]. In the agricultural environmental governance, the ecological restoration mechanism needs to fully clarify the responsible person of the pollution subject, the method of pollution compensation and the technology used for environmental governance.

Through questionnaire survey and regional agricultural economic literature survey, this study has grasped the economic structure type and environmental pollution characteristics of heavy metal pollution areas. The main pollution elements are As, Cd and Zn, among which Cd element pollution is relatively serious, accounting for 26.4% of farmland pollution detection. Through further investigation of the main pollution factors, pollution levels and pollution sources in the polluted areas, the pollution subjects such as sewage enterprises, pesticide and fertilizer users were identified. Based on the evolutionary game theory, some studies have developed a set of mathematical models to evaluate the attitude and preference to the ecological compensation plan. The three main stakeholders include farmers, local governments and business groups to investigate whether the asymptotic stability strategy of stakeholders can be realized, and the simulation analysis shows the sensitivity characteristics and evolution process of stakeholders affected by various factors. The results show that the threshold effect of these factors is an important basis for formulating the ecological compensation plan of the forest ecotourism system, and put forward the three-stage strategy and policy implications for the development of the operational ecological compensation plan of the forest ecotourism system [40]. According to the environmental compensation standard, the polluter will bear the main pollution control cost and agricultural loss cost, and determine the amount of compensation for farmland with different pollution levels according to the pollution level and scope. At the same time, in the operation of the compensation mechanism, local governments and enterprises need to actively participate in the environmental governance and recovery process, and select appropriate governance measures according to the type of pollution, so as to effectively improve the rural pollution problem and improve the application effect of the ecological compensation mechanism.

## 5. Conclusions

Environmental pollution has always been a major problem facing the development of modern industry. With the development of modern industrial technology, the development of green and new energy industry has become one of the important means to promote China's economic development. Heavy metal pollution in farmland was analyzed in this paper. In order to improve the quality of farmland polluted by heavy metals, a soil environmental evaluation system was constructed based on the restoration ecological compensation theory. The main pollution sources and pollution degree of heavy metal polluted farmland were analyzed, and the compensation plan for heavy metal polluted farmland was obtained. In addition, this paper selected a region of heavy metal polluted farmland as the research object; the types of heavy metals in farmland were tested and analyzed. The results showed that Cd was the main contaminant, accounting for 26.4%, followed by Pb, accounting for 6.8%. According to the established compensation scheme, ecological treatment of farmland with different pollution degree was carried out, and the total amount of ecological compensation reached CNY 32.35 million. In recent years, China has strengthened environmental health supervision, effectively improving regional ecological environment and air quality. However, some underdeveloped agricultural areas still face major environmental problems, such as heavy metal pollution and high concentration of atmospheric dust. Therefore, it is necessary to establish the corresponding system in environmental governance and strengthen people's attention to environmental health, so as to ensure the health of the ecological environment. However, there are

still some deficiencies in the study, which does not take into account the harm of heavy metal pollution to human physical and mental health, and the research needs further improvement in the future.

**Author Contributions:** Investigation, Z.C.; Writing—original draft, Z.C.; Supervision, X.Y. All authors have read and agreed to the published version of the manuscript.

**Funding:** This work was sponsored in part by Heilongjiang Provincial Research Plan for Philosophy and Social Sciences (No. 21JYC244).

**Institutional Review Board Statement:** Not applicable.

**Informed Consent Statement:** Not applicable.

**Data Availability Statement:** All data generated or analysed during this study are included in this published article.

**Conflicts of Interest:** The authors declare no conflict of interests.

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
