# Peer review of "Research on Restoration of Heavy Metal Contaminated Farmland Based on Restoration Ecological Compensation Mechanism"

_sustainability, doi:10.3390/su15065210_

Round 1
Reviewer 1 Report
Contamination of agricultural soils with heavy metals is a modern problem in agriculture. This is an issue that most needs new and better solutions, which the authors emphasize in abstract. At this point, there is already my first remark related to the fact that the introduction chapter, which is based on 5 items of the cited literature, should be expanded. In my opinion, this is definitely not enough and this chapter will not provide a proper introduction to the subject. And the topic is very common in the literature, and more evidence should be provided in the form of citations.
Chapter two - in my opinion the title of the chapter is incorrect and should be corrected. Table 1 has the wrong caption in my opinion. How can a table be a definition?
Chapter 3 - at the beginning there is a sentence about the subject of research presented in the article. If the article is not a review but only about research/experiment then it is completely wrongly constructed. In this case, there should be a material and methods chapter, etc. Even if it concerns a specific case study, it cannot stay in this form.
In the next part, solutions, proposed changes, models of solutions for the presented problem and several citations of literature referring to the topic are presented.
In my opinion, the article should be strongly reworded and improved. The structure itself leaves much to be desired. Another thing is the number of cited literature is 40 items. What the author had in mind? review article or result article? in both cases, it is essential to expand the citations. An original scientific paper cannot be based on only 40 items with such a common topic of the article. There should also be a discussion section, either separately or combined with previous chapters.
Author Response
- Contamination of agricultural soils with heavy metals is a modern problem in agriculture. This is an issue that most needs new and better solutions, which the authors emphasize in abstract. At this point, there is already my first remark related to the fact that the introduction chapter, which is based on 5 items of the cited literature, should be expanded. In my opinion, this is definitely not enough and this chapter will not provide a proper introduction to the subject. And the topic is very common in the literature, and more evidence should be provided in the form of citations.
Answer: Thank you very much for your kind comments. We provided more evidence in the form of citations in the revised paper. Such as:
According to the experience of ecological protection at home and abroad, it is found that harnessing rural ecological problems based on the ecological compensation mechanism is an effective way to improve the farmland environment and increase agricultural crop yield. At the same time, due to the excessive concentration of metal elements in metal-polluted soil, the farmland soil has problems of drought and high salinity, which inhibit the growth of plants. At present, the main remediation technologies of heavy metal contaminated soil are physical and chemical remediation technology, microbial remediation technology, ecological remediation technology, etc.
- Chapter two - in my opinion the title of the chapter is incorrect and should be corrected. Table 1 has the wrong caption in my opinion. How can a table be a definition?
Answer: Thank you very much for your kind comments. The title of Chapter two and Table 1 were modified as “2 Ecological compensation theory research and heavy metal farmland pollution grade evaluation” and “Assessment of soil pollution level”, respectively. In addition, The title of Sections 2.1 and 2.2, Table 2, Table 3 and Figures 1-3 were checked and optimized.
- Chapter 3 - at the beginning there is a sentence about the subject of research presented in the article. If the article is not a review but only about research/experiment then it is completely wrongly constructed. In this case, there should be a material and methods chapter, etc. Even if it concerns a specific case study, it cannot stay in this form.
Answer: Thank you very much for your kind comments. According to your kind suggestion, in section 3.1, we mainly discussed the methods and materials of the user compensation survey to make this part of the study more sufficient. Such as the first and second paragraphs of 3.1. As follows:
The compensation survey of users in heavy metal polluted farmland areas is mainly carried out in the form of questionnaire survey, field visit and consulting regional agricultural economic documents. Therefore, 276 questionnaires on heavy metal pollution compensation were distributed in the region, including the economic type of the local people, the understanding of heavy metal pollution, and the compensation provisions for heavy metal pollution. Finally, 265 copies were recovered, with a recovery rate of 96.01%.
According to the survey of regional visits, people in heavy metal pollution areas mainly rely on agricultural economy. At the same time, many enterprises choose to build factories and put into production in rural areas. Although it has driven the economic development of the region in the short term, in the long run, the agricultural ecology of some regions has been affected by industrial pollution, resulting in some cultivated farmland becoming heavy metal contaminated farmland
- In the next part, solutions, proposed changes, models of solutions for the presented problem and several citations of literature referring to the topic are presented.
Answer: Thank you very much for your kind comments. Solutions, proposed changes, models of solutions for the presented problem were revised and marked as red color.
Some references have been cited, such as references 28, 30 and 36.
- In my opinion, the article should be strongly reworded and improved. The structure itself leaves much to be desired. Another thing is the number of cited literature is 40 items. What the author had in mind? review article or result article? in both cases, it is essential to expand the citations. An original scientific paper cannot be based on only 40 items with such a common topic of the article. There should also be a discussion section, either separately or combined with previous chapters.
Answer:Thank you very much for your kind comments. The structure of the manuscript has been revised to meet the structural requirements. The article quotes 40 references, which are the number of references required for this research. The reference scope was modified to make the application more standardized. At the same time, according to your suggestion, a separate discussion section was added in section 4.3 to discuss the relevant value of the research content.
Reviewer 2 Report
Major corrections are required for the publication of this manuscript entitled “Research on restoration of heavy metal contaminated farmland based on restoration ecological compensation mechanism”.
- Abstract: This section doesn't highlight enough the novelty of this described manuscript. The exact applications of the study findings are also not emphasized enough which is insufficient to frame the whole picture of the present study.
- Major structural changes are required and paragraphs are in general too long, so the authors should reorganize it in a better way.
- Improve the introduction section with latest references. How heavy metal pollution can be reduced described other methods also. Like by using nanotechnology and microbial inoculants etc. Authors can consider these works: DOI 10.3389/fpls.2022.930340, https://doi.org/10.1007/s41742-022-00439-0
- The authors should address the goals of the study and emphasize research gaps, What should future studies in this area focus on? Please give some future work directions.
- Please rewrite the highlights or focus on that aspect more throughout the manuscript.
- The abbreviations should be written in full form in the first place. Please revise through the manuscript.
- Authors are advised to improve the quality of pictures.
- The discussion presented is very weak no strong comparison has been made with the literature to support the authenticity of the obtained results. Therefore, the authors are suggested to discuss their results with the following recent researches
Author Response
Major corrections are required for the publication of this manuscript entitled “Research on restoration of heavy metal contaminated farmland based on restoration ecological compensation mechanism”.
- Abstract: This section doesn't highlight enough the novelty of this described manuscript. The exact applications of the study findings are also not emphasized enough which is insufficient to frame the whole picture of the present study.
Answer: Thank you very much for your kind comments. We have highlighted the novelty and results of this study in the abstract in order to summarize the whole picture of this study.
- Major structural changes are required and paragraphs are in general too long, so the authors should reorganize it in a better way.
Answer: Thank you very much for your kind comments. According to your suggestions, the structure of the method part has been optimized, such as the re-segmentation of the method part 1 and the optimization of some details. See 2.1 and 2.2 for details, and mark them in red.
- Improve the introduction section with latest references. How heavy metal pollution can be reduced described other methods also. Like by using nanotechnology and microbialinoculants etc. Authors can consider these works: DOI 10.3389/fpls.2022.930340, https://doi.org/10.1007/s41742-022-00439-0
Answer: According to your kind suggestions, we rearrange the introduction and put the latest technology you provided for reference repair into it, such as the red-marked paragraphs 7, 8, 9, 10, etc. of the introduction. Thank you for your support.
- The authors should address the goals of the study and emphasize research gaps, What should future studies in this area focus on? Please give some future work directions.
Answer: Thank you very much for your kind comments. We added the direction and development trend that future research in this field should focus on. Such as:
In recent years, China has strengthened the development requirements for rural environment and food safety, making the agricultural ecological restoration compensation mechanism gradually perfect. In the ecological compensation mechanism, the main compensation requirement of "who pollutes, who compensates" is clearly stipulated. At the same time, with the gradual improvement of relevant environmental regulatory requirements and the innovation of heavy metal pollution remediation technology, the problem of heavy metal pollution in rural areas has also been alleviated[39]. In the agricultural environmental governance, the ecological restoration mechanism needs to fully clarify the responsible person of the pollution subject, the method of pollution compensation and the technology used for environmental governance. In the study, the types of economic structure and the characteristics of environmental pollution in heavy metal polluted areas were mastered through questionnaires and regional agricultural economic literature surveys, providing reference for the follow-up application of ecological compensation mechanism. At the same time, through further investigation of the main elements, pollution levels and pollution sources of pollution in the polluted areas, the pollution subjects, such as sewage enterprises, pesticide and fertilizer users, were identified[40]. According to the standard of environmental compensation, the polluter will bear the main pollution control costs, agricultural loss costs, etc. According to the type of pollution, local governments and enterprises need to actively participate in the process of environmental treatment and restoration, and select appropriate treatment measures according to the type of pollution, so as to effectively improve the rural pollution problem and improve the application effect of ecological compensation mechanism.
However, there are still some deficiencies in the study, which does not take into account the harm of heavy metal pollution to human physical and mental health, and needs further improvement in the future.
- Please rewrite the highlights or focus on that aspect more throughout the manuscript.
Answer: Thank you very much for your kind comments. The focus of the whole article is to grasp the characteristics of pollution elements in the polluted area and build a compensation mechanism model based on the characteristics. The specific situation of pollution in the polluted area is obtained by building a model, and the amount of pollution compensation is obtained. At the same time, take unnecessary pollution control measures according to the characteristics of pollution.
- The abbreviations should be written in full form in the first place. Please revise through the manuscript.
Answer:According to your kind suggestions, the abbreviations of the full text are supplemented and marked with red, as shown in the full description of PH in Table 2. At the same time, in the first paragraph of 2.1, rows 6 and 7 are marked in red to represent the elements completely.
- Authors are advised to improve the quality of pictures.
Answer: Thank you very much for your kind comments. High resolution pictures were provided in the revised paper. In addition, some data can only be visualized in the form of tables. If the variable relationship of data cannot be visualized in the form of graphs, the understanding is also more complex. Some tables are presented in the form of three-line tables to improve the quality of tables, as shown in Table 5
- The discussion presented is very weak no strong comparison has been made with the literature to support the authenticity of the obtained results. Therefore, the authors are suggested to discuss their results with the following recent researches
Answer: Thank you very much for your kind comments. The discussion section is enhanced by a strong comparison with the literature to support the authenticity of the results obtained. In addition, unlike traditional technical manuscripts, ecological compensation mechanism is a common ecological governance strategy. The research is mainly to reflect the effect of ecological governance and the final cost, and cannot reflect the cost optimization through multiple technologies. The compensation cost is mainly obtained through technology, and the heavy metal pollution problem is treated according to the research results.
Reviewer 3 Report
The manuscript describes the ecological restoration compensation mechanism for restoration of farmland polluted with heavy metals. Paper is based, among others, on the case study. Authors stated that ecology of the studied area has improved very clearly due to the application of different ecological restoration schemes. Title is adequate for the content of paper. The structure of the manuscript is atypical, since sections such as results and discussion are not included. I think the point 4.2 could be a part of conclusions.
Chemical analysis, without any statistics, is the weakest part of the work.
There are some minor corrections that should be made:
page 9
using precision instruments - what excactly it means ?
lightly polluted, moderately polluted and heavily polluted farmland - better use the terms from the table 1
quality safety labels - not levels ??? please check it
environmental protection safety standard - maybe you can provide some regulations ???
page 10
As shown in Figure 4(b), the longest restoration time is 40 months for severely polluted farmland while the shortest is 15 months for mildly clean farmland - better use the terms from the table 1
Table 2 - Hg - Mercury; Ph – pH; ph – pH; ”–” in the column metal substances - what does it mean ???
Table 3 - x1 _ … X 6 - I think factors should be unified
Table 4, Figure 4 - Cost (yuan/mu) - what is ”mu” ?
Table 2, 5 - Heavy metal substance - ”Heavy metal” is better
Figure 1 - please correct ”Farming” and ”Working”
Figure 2 - Main Sources of Farmland Heavy Metals - better in lower case: Main sources of farmland heavy metals
Figure 3 - Figure 3. Experimental results of heavy metal farmland pollution sources - better without ”sources” … because the term ”sources” means something else (it is shown on the Figure 2); 3a - Number of pollution source measuring points (X-axis) – better like this: Number of pollution measuring points; 3b – better, for example: ”Percentage of pollution measuring points for each metal” (X-axis)
Figure 4b - Pollution control days/month - days ??? In the text only months are mentioned
Equation 3, 4 – What values can ”n” take ?
Equation 1, 5, 6 - Are all the variables unitless ? For example Ci in equation 1…
Author Response
- The manuscript describes the ecological restoration compensation mechanism for restoration of farmland polluted with heavy metals. Paper is based, among others, on the case study. Authors stated that ecology of the studied area has improved very clearly due to the application of different ecological restoration schemes. Title is adequate for the content of paper. The structure of the manuscript is atypical, since sections such as results and discussion are not included. I think the point 4.2 could be a part of conclusions.
Answer: Thank you very much for your kind comments. The treatment of heavy metal pollution in farmland is based on conclusions and specific pollution treatment opinions are adopted according to the specific situation of pollution. The text of this part was reorganized to ensure the compactness of the manuscript.
- Chemical analysis, without any statistics, is the weakest part of the work.
Answer: Thank you very much for your kind comments. We have discussed it in the section 4.1 in the revised paper as follows:
A chemical enterprise in the rural area failed to discharge industrial production sewage into the rural river channel in accordance with the regulations during the production operation. The people in this area used the polluted water source for farmland irrigation, spraying and fertilization without knowing, resulting in heavy metal pollution of a large number of farmland in this area. The soil samples from 265 typical polluted places in the polluted area were collected, and the impurities were filtered, and the content of heavy metal elements in the soil was detected by atomic fluorescence spectrometry. According to the inspection of the expert team, the contaminated farmland in this area is as high as 34800 mu, among which the contaminated farmland is classified into 30000 mu, 3200 mu relatively clean, 800 mu slightly, 500 mu moderately and 300 mu seriously. It can be seen that the heavy metal pollution of farmland in this area is relatively serious. The experimental analysis of heavy metal properties of polluted farmland is shown in Figure 3.
- There are some minor corrections that should be made:
(1) page 9
using precision instruments - what exactly it means ?
Answer: Thank you very much for your kind comments. “precision instruments” was deleted in the revised paper.
lightly polluted, moderately polluted and heavily polluted farmland - better use the terms from the table 1
Answer:Thank you very much for your kind comments.It has been modified according to the standards in Table 1.
quality safety labels - not levels ??? please check it
Answer: According to your kind suggestion, “quality safety labels” was changed to “quality safety level”.
environmental protection safety standard - maybe you can provide some regulations ???
Answer: According to your kind suggestion, In the 328-329 lines of the article, the specific environmental protection and safety standards are pointed out.
(2) page 10
As shown in Figure 4(b), the longest restoration time is 40 months for severely polluted farmland while the shortest is 15 months for mildly clean farmland - better use the terms from the table 1
Answer: Thank you very much for your kind comments. The incorrect "severly polluted farmland, and milly clean farmland" statements have been replaced with the standard names in Table 1.
Table 2 - Hg - Mercury; Ph – pH; ph – pH; ”–” in the column metal substances - what does it mean ???
Answer: Thank you very much for your kind comments.The "–"in Table 2 means there is no substance of this grade in conventional farmland. Now, "–" is deleted in the revised paper.
Table 3 - x1 _ … X 6 - I think factors should be unified
Answer: Thank you very much for your kind comments. Now, they were unified in Table 3 in the revised paper.
Table 4, Figure 4 - Cost (yuan/mu)- what is ”mu” ?
A: Thank you very much for your kind comments. The unit of the full text is acres, and all are modified. Thank you for your comments.
Table 2, 5 - Heavy metal substance - ”Heavy metal” is better
Answer: Thank you very much for your kind comments.“Heavy metal substance” was changed to “Heavy metal” in Table2 and 5.
(3) Figure 1 - please correct ”Farming” and ”Working”
Answer: Thank you very much for your kind comments.“Farming” and “Working” were changed to “Farmer” and “Worker”.
Figure 2 - Main Sources of Farmland Heavy Metals - better in lower case: Main sources of farmland heavy metals
Answer: Thank you very much for your kind comments.“Main Sources of Farmland Heavy Metals” was changed to “Main sources of farmland heavy metals”.
Figure 3 - Figure 3. Experimental results of heavy metal farmland pollution sources - better without ”sources” … because the term ”sources” means something else (it is shown on the Figure 2); 3a - Number of pollution source measuring points (X-axis) – better like this: Number of pollution measuring points; 3b – better, for example: ”Percentage of pollution measuring points for each metal” (X-axis)
Answer: Thank you very much for your kind comments.“sources” was changed to “points”. 3a - Number of pollution source measuring points (X-axis) was changed to “Number of pollution measuring points”. 3b – Y-axis was changed to “Percentage of pollution measuring points for each metal”.
Figure 4b - Pollution control days/month - days ??? In the text only months are mentioned
Answer: Thank you very much for your kind comments.Figure 4b– X-axis was changed to “Pollution control period/month”.
Equation 3, 4 – What values can ”n” take ?
Answer: Thank you very much for your kind comments.“n”is the factor serial number.
Equation 1, 5, 6 - Are all the variables unitless ? For example Ci in equation 1…
Answer: Thank you very much for your kind comments.The variable values in equations 1, 5 and 6 are mainly used to build the indicator model. Generally, the unit is not given but the meaning of the variable needs to be explained. Ci is the opportunity cost.
Round 2
Reviewer 1 Report
Thank you for making the corrections.
In my opinion, however, the discussion section should be improved once again, as it is based on a few literal citations of literature. Given the size of the research results chapter, the discussion chapter leaves much to be desired.
Author Response
In my opinion, however, the discussion section should be improved once again, as it is based on a few literal citations of literature. Given the size of the research results chapter, the discussion chapter leaves much to be desired.
Reply: The discussion section has been improved carefully.
Heavy metal pollution in rural areas is a hot topic of social concern. Heavy metal pollution has a serious impact on environmental ecology, modern agriculture and food safety. Applying the ecological compensation mechanism to the environmental treatment process of heavy metal polluted areas provides a new direction for the remediation of heavy metal polluted farmland and the development of agricultural economy. Heavy metal pollution control in rural areas has been criticized for a long time[36]. The main reason is that the heavy metal pollution problem in rural areas can not clarify the responsibility of the pollution subject, the pollution compensation method, the pollution object and the pollution evaluation standard. At the same time, a survey was carried out on the development structure of rural economy in heavy metal pollution areas[37]. Most of the villagers in the remaining villages mainly rely on agricultural economy, while a small number of villagers choose to work in nearby village enterprises. This economic structure led to the local people's lack of awareness of the hazards of heavy metal pollution, and even many people were employed in sewage enterprises, resulting in the phenomenon of shelter and illegal pollution, which affected the restoration of heavy metal contaminated farmland[38].
In recent years, China has strengthened the development requirements for rural environment and food safety, making the agricultural ecological restoration compensa-tion mechanism gradually perfect. In the ecological compensation mechanism, the main compensation requirement of "who pollutes, who compensates" is clearly stipulated. At the same time, with the gradual improvement of relevant environmental regulatory re-quirements and the innovation of heavy metal pollution remediation technology, the problem of heavy metal pollution in rural areas has also been alleviated[39]. In the agri-cultural environmental governance, the ecological restoration mechanism needs to fully clarify the responsible person of the pollution subject, the method of pollution compensation and the technology used for environmental governance.
Through questionnaire survey and regional agricultural economic literature survey, this study has grasped the economic structure type and environmental pollution characteristics of heavy metal pollution areas. The main pollution elements are As, Cd and Zn, among which Cd element pollution is relatively serious, accounting for 26.4% of farmland pollution detection. Through further investigation of the main pollution factors, pollution levels and pollution sources in the polluted areas, the pollution subjects such as sewage enterprises, pesticide and fertilizer users were identified. Based on the evolutionary game theory, some studies have developed a set of mathematical models to evaluate the attitude and preference to the ecological compensation plan. The three main stakeholders include farmers, local governments and business groups to investigate whether the asymptotic stability strategy of stakeholders can be realized, and the simulation analysis shows the sensitivity characteristics and evolution process of stakeholders affected by various factors. The results show that the threshold effect of these factors is an important basis for formulating the ecological compensation plan of the forest ecotourism system, and put forward the three-stage strategy and policy implications for the development of the operational ecological compensation plan of the forest ecotourism system[40].According to the environmental compensation standard, the polluter will bear the main pollution control cost and agricultural loss cost, and determine the amount of compensation for farmland with different pollution levels according to the pollution level and scope. At the same time, in the operation of the compensation mechanism, local governments and enterprises need to actively participate in the environmental governance and recovery process, and select appropriate governance measures according to the type of pollution, so as to effectively improve the rural pollution problem and improve the application effect of the ecological compensation mechanism.
Reviewer 2 Report
The content of this work is inadequate to the publication in MDPI journal.
Author Response
The English language and style have been fine/minor spell checked.
And the content have been improved.